# Plant-Based Meat Proteins: Processing, Nutrition Composition, and Future Prospects

**DOI:** 10.3390/foods12224180

**Published:** 2023-11-20

**Authors:** Jialing Yu, Liyuan Wang, Zhaowei Zhang

**Affiliations:** 1College of Food Science and Nutrition, University of Leeds, Leeds LS2 9JT, UK; yujialing951@gmail.com; 2Key Laboratory of Biology and Genetic Improvement of Oil Crops, Oil Crops Research Institute of Chinese Academy of Agricultural Sciences, Wuhan 430062, China; 18534369137@163.com; 3Hubei Hongshan Laboratory, Wuhan 430070, China

**Keywords:** plant-based meat, protein alternative, nutrition, extraction, food processing

## Abstract

The growing need for plant-based meat alternatives promotes the rapid progress of the food industry. Processing methods employed in plant-based meat production are critical to preserving and enhancing their nutritional content and health benefits, directly impacting consumer acceptance. Unlike animal-based food processing, the efficiency of protein extraction and processing methods plays a crucial role in preserving and enriching the nutritional content and properties. To better understand the factors and mechanisms affecting nutrient composition during plant-based meat processing and identify key processing steps and control points, this work describes methods for extracting proteins from plants and processing techniques for plant-based products. We investigate the role of nutrients and changes in the nutrients during plant protein product processing. This article discusses current challenges and prospects.

## 1. Introduction

The global meat demand has increased by 58% over the past two decades due to the increasing global population and rapid economic development [1,2,3]. Growing animal meat production requests will consume more fresh water and farmland, affecting the ecosystem and environment by emitting overloaded greenhouse gases. In this case, the production of animal meat, especially livestock meat, cannot meet the status quo. Thus, developing a promising alternative to meat is highly requested. The emerging plant-based meat is one promising way out. Plant-based meat has several names, like meat analogues, artificial meat, cultured meat, vegan substitutes, and meat substitutes [4]. Typically, this meat originates from various plants, like soybean, rice, wheat, oats, peas, and lentils [5]. With improved taste/flavour, colour, texture, nutritional value, low price, and diversity, plant-based meat attracts increasing consumers and covers a growing market. Plant-based meat demonstrates that it is a promising candidate for human health. Plant-based meat contains proteins, amino acids, texture, minor elements, bioactive compounds, calcium, iron, protein, fatty acids, and minerals.

Processing is essential for preserving and enhancing its nutritional composition and health effects, directly influencing consumer acceptance [6]. Nowadays, many investigations focus on meat alternatives, including the current situation and challenges of plant-based and cell-based meat alternatives regarding production efficiency, product characteristics and impact categories [7], and regional assessment of plant-based alternatives to meat consumption [7]. Processing plant-based meat involves protein extraction and processing methods. Extraction methods include chemical, enzyme-assisted, and physical-assisted techniques [8,9]. Processing techniques encompass extrusion processing, 3D printing, and electrospinning methods. Unlike animal food processing, processing plant-based protein substitutes requires the following considerations. One is the satisfied optimization process for efficient, economical, and green purposes. Furthermore, proper proteins should transform into plant proteins with equivalent characteristics to animal proteins. Third, the extraction and processing methods must preserve and enrich the nutrients.

A systematic review of the processing of plant-based proteins is requested to elucidate the migration and transformation patterns of nutritional components during processing, providing advanced information for producing high-quality plant protein products in the future. Several experts have published excellent reviews on plant-based meat, covering methods for preparing plant-based protein products and functional and sensory characteristics of plant-based milk [9,10]. These reviews have provided valuable scientific foundations for future research on plant-based meat. To the best of our knowledge, few studies focused on the impact of nutritional components during the processing of plant-based meat. We aim to gain a deeper understanding of the factors and mechanisms influencing the nutritional components during the processing of plant-based meat and to identify key processing steps and control points for enhancing nutritional content. Concerning plant-based meat, this work begins by introducing methods for protein extraction from plants and processing techniques, then it summarizes the role of nutritional components products, examines the variations in nutritional content during the processing of plant-based products, and finally, it discusses challenges and prospects. By integrating disciplines such as nutrition, food processing, and analytical chemistry, this work aims to identify critical control points for understanding the changes in nutritional components during the processing of plant-based meat.

## 2. Processing Method

### 2.1. Preparation of Plant-Based Alternative Protein

#### 2.1.1. Chemical Method

The chemical method is a traditional method of extracting vegetable protein, including water extraction, alkali extraction, acid extraction, and organic solvent extraction according to different extraction solvents (Table 1). Water extraction is water-based extraction using salt (NaCl), sodium dodecyl sulfonate (SDS), and non-ionic detergent (NP-40 and TritonX100) in hot or freezing water [11,12]. This method is inefficient for extracting some plant proteins, such as proteins rich in hydrophobic amino acid residues (leucine, isoleucine, etc.), due to their low solubility in water. Alkaline extraction is the most used method in industrial plant protein extraction [13], which increases the pH (pH = 9–12) by adding NaOH or KOH to the solution to keep it away from the isoelectric point of plant protein (pH = 4~6.5) and increase the protein solubility. Protein extraction yields are higher in alkaline environments since the alkaline pH results in the breakdown of protein disulfide bonds (e.g., soya protein, pea protein) [14]. The acid extraction method changes the charge of the protein by lowering the pH of the solution, keeping it away from the isoelectric point of the vegetable protein. To ensure the extraction efficiency of plant protein, use of the alkaline extraction method for acidic proteins and the acidic extraction method for essential proteins is necessary. In addition, lipid-binding proteins and proteins with many nonpolar side chains are easily soluble in organic solvents (ethanol, butanol, acetone, etc.) but insoluble in water, dilute salt solution, dilute acid, or dilute alkali. One can extract these proteins with organic solvents, as they exhibit lipid binding ability, possess non-polar or polar side chains, and contain aromatic amino acids [13].

#### 2.1.2. Enzyme-Assisted Method

Enzyme-assisted extraction (EAE) is a widely employed method in the commercial production of plant proteins (e.g., pectinase, glucoamylase, cellulase) [12] (Advances in the plant protein extraction: Mechanism and recommendations), enabling the recovery of high-quality proteins from sources such as grain and oilseed residues [15]. EAE leverages enzymes to disrupt the structure and integrity of cell walls, facilitating the separation and extraction of proteins (as shown in Figure 1A) [16]. Research by Perovic et al. [17] demonstrated that the use of enzyme complexes in combination with traditional alkaline extraction resulted in a 21% increase in protein yield compared to alkaline extraction alone for soybean flour. Furthermore, pretreatment with enzymes such as xylanase, pectinase, and cellulase led to a 13% increase in protein yield. EAE can be combined with other techniques, such as ultrasound and microwave, to enhance protein extraction efficiency further [18]. EAE represents an effective and environmentally friendly method for plant protein extraction. However, it does have certain drawbacks, including a slow extraction speed, higher operational costs, challenges in scaling up for large-scale extraction, and inconsistent yields [12].

#### 2.1.3. Physical Assisted Method

Physical-assisted methods encompass high pressure, pulsed electric field, ultrasonic-assisted extraction (UAE), microwave-assisted extraction, and other techniques. These methods operate by disrupting the cell walls, facilitating liquid penetration into the intracellular space, thereby dissolving and releasing proteins into the liquid medium [19]. This approach enhances the efficiency of plant protein extraction and improves protein modifications [20]. Dong et al. [21] demonstrated that ultrasonic-assisted extraction, compared to traditional alkaline extraction, increased the protein yield from rapeseed cake, with a protein recovery rate of 76.83%. In contrast, the conventional method’s protein recovery rate was only 35.43%. This highlights the significant improvement in protein extraction efficiency achieved through optimized ultrasonic-assisted extraction methods. Similarly, Lafarga et al. [22] employed UAE to extract proteins from Ganxet beans, achieving a protein recovery rate of 78.73%, representing a remarkable increase of 37.98% compared to previous experiments.

**Table 1 foods-12-04180-t001:** Examples of processing methods.

Target	Extraction Method	Extraction Reagents	Extraction Efficiency	Refs.
Moringa oleifera	Organic solvent extraction	PE-ethanol/water	recovery rate: 33%	[23]
Mung bean protein	Alkaline extraction	KOH solution	77.32% protein	[24]
Rapeseed	Enzyme-assisted method	Protease A-01	80.7% (cold-pressed rapeseed meal), 78.3% (pre-pressed rapeseed meal)	[25]
Potato Proteins	Enzyme-assisted method	α-amylase, galactanase	86.5% (protein content)	[26]
Flaxseed	Enzyme-assisted method	Cellulase	86.80 ± 0.91% (enzymatic-solvent flaxseed protein concentrates, ES-FPC)	[15]
Soybean	Reverse micelles extraction	25 min, pH 3.0, 45 °C	48.66%	[27]
Mango peel pectin	Microwave-assisted extraction	700 W, 3 min	/	[28]
Sesame Bran	Ultrasound-assisted extraction	Water, 35 kHz, 18 time	59.8%	[29]
Chlorella vulgaris (SAG 211-12)	Pulsed electric field (PEF) treatment	1.94 kJ kg sus ^−1^	96.6 ± 4.8% (available free protein)	[30]
Wet biomass of Arthrospira platensis (spirulina)	High-pressure assisted extraction	300 MPa, 20 °C, 10 min	90% (C-phycocyanin), 60% (total soluble proteins)	[31]

### 2.2. Processing Technology of Plant-Based Alternative Protein Products

#### 2.2.1. Extrusion Technology

Extrusion technology involves using pressure from an extruder to force food materials through a die, transforming plant proteins into structured aggregates or fibres, thereby altering their shape and structure within a multifunctional system [4]. Depending on the moisture content, the extrusion can be categorized into low-moisture extrusion (<30%) and high-moisture extrusion (50%) [23]. The former is widely applied in non-meat products, such as the production of textured vegetable protein (TVP), where a high protein content in the raw material is not a prerequisite [24,25]. TVP serves as an ingredient in various food applications, including taco fillings and pizzas, owing to its texture and protein content, making it an ideal substitute for animal-derived proteins [4]. Despite the technology’s broad applicability, high maturity, and low equipment costs associated with low-moisture extrusion, products produced through this process require rehydration before use [26,27].

High-moisture extrusion is a process technology used to replicate conventional whole-muscle meat’s appearance and fibrous texture, commonly employed for soybean, wheat, peas, and other ingredients [28]. Under the influence of heat and shear forces, high-moisture extrusion induces structural changes in the major protein, such as soy protein isolate (SPI), within extruded meat analogues. Low-friction and low-viscosity twin-screw extrusions are considered optimal choices for high-moisture protein mixtures [29,30]. Compared to low-moisture extrusion, high-moisture extrusion better replicates the texture of meat products and is suitable for whole-muscle meat analogues [31]. This is because, in a high-moisture environment, the presence of water facilitates the formation of a tighter network structure of proteins and fibres during the extrusion process, thereby increasing the chewiness of the extruded meat analogues and making them more similar in texture to meat products [29]. For example, Cho et al. [29] found that using high-moisture extrusion technology for oyster mushrooms improved the chewiness and cutting strength of the SPI-based meat analogue.

#### 2.2.2. Three-Dimensional Printing

Three-dimensional (3D) printing creates a three-dimensional object layer-by-layer using a computer-created design, which can reduce material wastage [32,33]. In food research, 3D printing technology primarily encompasses four types: extrusion, selective sintering, binder jetting, and inkjet printing [32]. Three-dimensional printing allows for delicate structures for plant-based protein meat, achieving a texture like real meat. It also enables the customization of protein-based meat’s shape, flavour, colour, and nutritional composition, offering advantages such as design flexibility, high precision, and minimal waste generation [34,35]. Cross-linking and pretreatment modifications are necessary for plant-based proteins to facilitate smooth extrusion and achieve sufficient mechanical strength in plant-based protein-based printing materials [36]. Shahbazi et al. [37] utilized 3D printing technology to create a plant-based meat substitute using soy protein isolates, capitalizing on their high water-holding capacity and gelation properties. Additionally, Dutta et al. [38] employed edible polysaccharides (alginate) and proteins (gelatin) derived from natural sources to proliferate bovine skeletal muscle cells (bMSCs) in a low-serum environment to develop fat-free meat using bioprinting techniques.

#### 2.2.3. Emerging Processing Methods

Electrospinning is a method to produce ultrafine (in nanometers) fibres by charging and ejecting a polymer melt or solution through a spinneret under a high-voltage electric field and to solidify or coagulate it to form a filament, like the extrusion technology [39]. It will be affected by globulin, but mixing high-concentration soy protein isolate and sodium alginate can improve the spinnability of the spinning solution, avoid harmful reactants, and improve the meat fibre quality [39]. Wet spinning, conversely, is a processing method that involves spinning polymer solutions or melts into fibres through immersion or extrusion, as shown in Figure 2II. It is suitable for producing micro- and nanoscale fibres. However, its application in food is limited due to the use of organic solvents, chemical crosslinking agents, and synthetic polymers during manufacturing, which are not ideal for food applications [40]. Therefore, expanding the application of wet spinning technology in food requires exploring environmentally friendly reactants and solvents. The Shear cell technology, invented by Professor Atze Jan van der Goot and his team at Wageningen University, reshapes polymeric structures based on stable shear flow fields. This low-energy technique has reached the sixth generation of high-temperature (>100 °C) and laboratory-scale equipment but has not been industrialized yet [5]. Protein alignment and fibre structure formation can be achieved through simple shear and heat. The Couette Cell has been utilized in this context to shear protein mixtures [41].

#### 2.2.4. Technology for Improved Functional Properties of Plant Proteins

The diverse sources of plant proteins lead to solubility, foaming, and emulsification challenges. Each plant protein possesses unique structures and properties, so multiple approaches are required to address these challenges (Figure 2I, Table 2). For example, the incomplete solubility of proteins may lead to uneven particles or texture issues within the plant protein meat product. Poor foaming properties can make the product more challenging during cooking, less porous, and have few juicy characteristics. Insufficient emulsification leads to the separation of fats from water, negatively impacting the overall texture and mouthfeel [10,42].

Various approaches to improve the quality and market acceptance of plant-based meat products include fibrillation, enzymatic hydrolysis, pH-shifting, heat treatment, additives, and microbial fermentation. However, these methods are usually accompanied by increased production costs. During fibrillation, mechanical shearing forces plant protein into smaller and more soluble units. This method improves plant protein meat’s foaming and emulsifying properties [43]. Enzymatic digestion can make large molecular structures of plant proteins into smaller fragments, thus improving proteins’ solubility and emulsification [10]. On the other hand, enzymatic digestion may alter the protein flavour or trigger bitterness, affecting the mouthfeel of the final meat product [44]. A pH value adjustment can alter the charge distribution of the meat protein, thereby affecting its three-dimensional structure and contributing to solubility and emulsification [45]. Heat treatment improves emulsification and foaming by precisely controlling the heating temperature and time. Careful control is suggested to maintain the taste and colour of the final product [46]. Food-grade auxiliaries, like xanthan gum or galactosidase, can significantly improve the solubility, emulsification, and stability of vegetable proteins [47]. Microbial fermentation uses microorganisms, such as yeasts or bacteria, to modify the structure and function of proteins. For example, lactic acid bacteria can form firm meat, while the modified yeast Pichia pastoris can produce soy leghemoglobin that resembles the flavour and colour of animal meat, thus improving the taste and appearance of plant-based burgers [48].

**Figure 2 foods-12-04180-f002:**
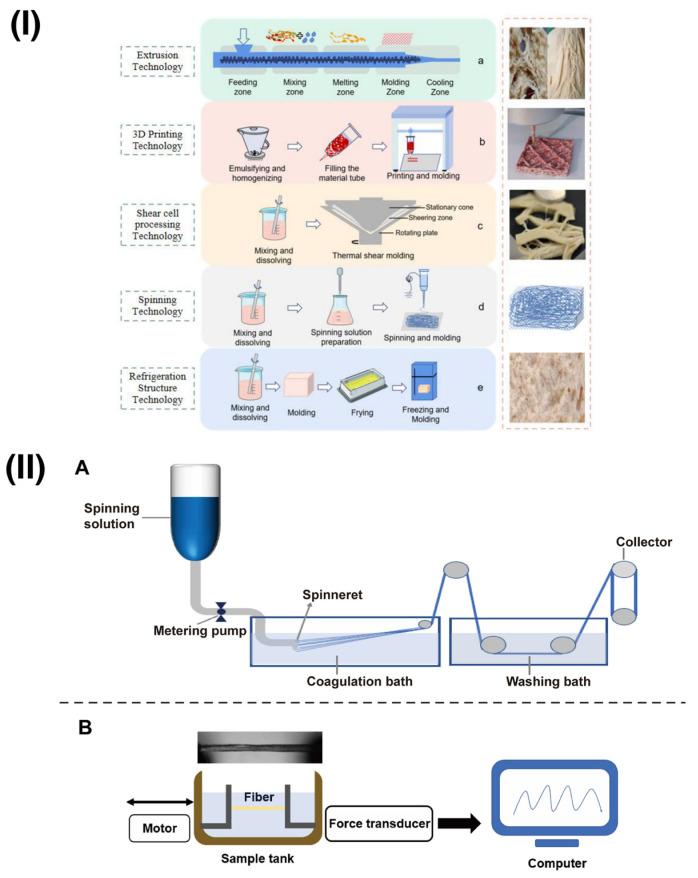
(**I**) The figure illustrates the processing principles, workflows, and final product outcomes for various moulding processes, including extrusion, 3D printing, shear cell, electrostatic spinning, and refrigeration structure technologies, highlighting how ingredients and processes influence the structural quality of recombinant plant-based meat alternatives and their components in a study. (**II**) Process flow diagram (**A**) of wet-spun SA/SPI composite fibres; schematic representations (**B**) of the fresh fibres’ tensile strength test [39,49].

**Table 2 foods-12-04180-t002:** The main types of processing methodology.

Processing Methodology	Advantages	Disadvantages	Refs.
Extrusion technology	(1) Directly simulating the whole cut; (2) Most economical; (3) Easy to operate and clean; (4) Saves water.	(1) Requires high temperature and pressure; (2) Needs the highest mechanical energy; (3) Difficult to modify the formulations and texture of products.	[2,23,50,51]
3D printing	(1) Easy to control the distribution of different components to obtain a more similar appearance and appealing shapes; (2) Easy to modify; (3) Small production space and ease of operation.	(1) Low extrusion force of 3D printer; (2) Limitations in printable plant materials; (3) High requirements on the flow characteristics of inks.	[34,52,53]
Electrospinning	(1) Uniformity of fibrils diameter; (2) Affordable cost; (3) Scalable production.	(1) Huge waste of water resources to wash the solid fibres; (2) Difficult to modify the formulations and texture of products; (3) Poor mouthfeel.	[39,54]
Shear cell	Requires lower mechanical energy than the extrusion technique.	(1) Requires high temperature and pressure; (2) Difficult to modify the formulations and texture of products.	[41,55]

## 3. Nutrition in Processing

### 3.1. Protein

Protein is an extraordinarily complex substance with great nutritional value for human essential chemical processes and a major component of plant-based meat fibres [56]. The main sources of plant-based proteins include beans, fruits and vegetables, grains, nuts, and seeds. Considering their availability, cost, and processing methods, soybeans, peas, and wheat gluten are widely utilized in plant-based protein products [57]. Diverse types of plant proteins possess varying structures and functionalities. For example, soybean protein exhibits better gelation properties compared to pea protein. Plant-based meat derived from soy protein is firmer and more elastic while lacking methionine than pea protein [58].

Processing steps will influence the functional properties, digestion and absorption, and physiological activities of proteins. During extrusion processing, protein denaturation occurs due to heat instability. Hydrophobic residues are exposed under high shear and temperature conditions, allowing proteins to unfold and align along the flow direction in the extruder. This process can form new intermolecular chemical bonds and aggregation in protein and change its content and properties [49,59]. The stability of high-moisture or low-moisture protein extrudates relies on hydrogen bonds/disulfide bonds and hydrophobic/disulfide bonds, respectively. Cooking temperature, screw speed, feed rate, and moisture content collectively affect the functionality of plant proteins after extrusion processing. Furthermore, extrusion processing at high temperatures can promote the Maillard reaction. This process can alter the protein structure, nutritional properties, and functional characteristics and lead to the formation of potentially harmful compounds [60]. The processing of plant-based meat includes homogenization, heating, and separation, wherein heating may modify the structure and interactions of plant proteins [61].

In fermentation, microorganisms’ life activities under aerobic or anaerobic conditions can prepare microbial cells, direct metabolites, or secondary metabolites. Fermentation can increase the protein content during the growth of food-grade microorganisms and improve amino acid composition [62]. After fermentation via plant Lactobacillus fermentation, sorghum protein (from 53% to 69%) and total protein content increased (from 47% to 69%) simultaneously [63]. Bacterial fermentation can increase the lysine content of fermented foods by producing peptides and amino acids via the microbial degradation of proteins [64].

### 3.2. Carbohydrates and Dietary Fiber

Carbohydrates, starch as a typical one, are the major nutrients in foods and drinks. Starch has two categories, which are amylose (linear) with viscidity and amylopectin (branched) with hardness and expansibility [8]. Common starches in plant-based meat are made from corn and wheat. Increasing the content of branched starch can effectively improve the fibre structure, stability, and hardness of the restructured products. Starch gelatinization can increase its viscosity initially but decreases its viscosity over a long time and exceeds shear forces. Starch gelatinization can affect the fibre structure, tensile property, and elasticity of plant-based meat while leading to changes in colour through the Maillard reaction [65]. Adding potato starch to plant-based meat improved the cohesion between various proteins, resulting in a compact structure and good juiciness [53]. Moreover, extrusion processing and 3D printing can induce starch gelatinization, smash starch particles, and weaken hydrogen bonds. During extrusion processing, using less starch particles leads to a low viscosity with increasing screw speed. Extrusion leads to the degradation of amylopectin and the reduction of sugars. After disrupting primary and secondary valence bonds and hydrogen bonds between adjacent starch molecules, the resulting porous and spongy product can increase the contact with amylase, thereby significantly enhancing the digestibility of starch [66,67,68].

Dietary fibre encompasses both soluble and insoluble fibres. Soluble dietary fibre can be used by beneficial microorganisms and regulate lipids, blood glucose, and weight. On the other hand, insoluble dietary fibre promotes gastrointestinal motility [69,70]. Adding dietary fibre enhances products’ nutritional value and the cohesiveness and texture of plant-based meat. For example, incorporating the dietary fibre concentrate from chicory forced roots into plant-based burgers resulted in better baking performance, nutritional value, hardness, shear force, and improved sensory attributes [71]. The green pea fibre increased the hardness and elasticity of chicken nuggets [72], while oat fibre increased beef mince’s gel-like properties and mechanical performance [73]. Upon high temperature, pressure, and shear force, extrusion processing disrupts glycosidic bonds in insoluble polysaccharides, converts polysaccharides to smaller soluble components, alters the proportion of dietary fibre, and thereby facilitates the digestion and absorption in plant-based meat [74]. During fermentation, enzyme-mediated reactions can break down plant cell walls and release phenolic compounds with a bounded form [75]. As 3D printing technology is in the preliminary stages, many researchers reported this technology in plant-based meat processing.

### 3.3. Lipids

Lipids maintain product tenderness, juiciness, texture, and flavour release and provide nutrition and energy [76,77]. Adding lipids to plant-based meat increases nutritional value, colour, tenderness, and structural integrity [78]. Lipids in plant-based meat products include olive, rapeseed, sesame, and coconut oil. Ran et al. [79] found that adding 3.5% sunflower seed oil to plant-based fish balls’ improved texture and water-holding capacity. One reason was the enhanced interaction between hydrophobic amino acids and the hydrocarbon side chains of plant oil, while another was the increased elasticity and enhanced water-holding capacity [80].

During extrusion, lipids impact the product’s texture, nutritional quality, and sensory attributes via emulsification and plasticization, imparting suitable texture and viscosity to the extruded products [81]. In the extrusion process, lipids can interact with proteins through covalent interactions, electrostatic interactions, hydrogen bonds, and van der Waals forces. Lipids can form complexes with starch. Additionally, lipids can inhibit the activity of lipases in the raw materials, preventing the generation of free fatty acids and enzymatic lipid oxidation [82]. Furthermore, the extrusion process influences the flavour of plant-based meat products. Lipids undergo oxidation and degradation under high-temperature and high-pressure conditions, forming volatile compounds (aldehydes, ketones, alcohols, and esters) as essential contributors to meat-like flavours. The lipid oxidation products can participate in the Maillard reaction to form meat-like flavours [5].

### 3.4. Vitamin and Mineral Salt

Plant-based meat can provide micronutrient profiles similar to animal meat by replacing ingredients and incorporating fortified nutrient additives [12]. Plant-based chicken-like nuggets using soy protein, mushrooms, and wheat bran provided lower lipid and sodium content and higher potassium, zinc, and iron than chicken nuggets [83]. Commercialized plant-based meat made from soy had higher dietary fibre, omega-3 fatty acids, and higher levels of trace elements such as iron, zinc, vitamin B1, vitamin B2, vitamin B6, and folate. On the other hand, plant-based meat products primarily made from nuts had higher fat content, monounsaturated fatty acids, and niacin levels.

During the extrusion process, barrel temperature, screw speed, moisture content of the raw materials, and die diameter affected the retention of vitamins in plant-based meat. The extrusion processing retained 44% to 62% of B vitamins, while thiamine had poor stability [84]. The lost vitamins during extrusion were vitamins A, E, C, B1, and folate, while vitamins B2, B6, B12, niacin, pantothenic acid, and biotin were stable [85]. Additionally, the extrusion processing increased the iron content in the resulting products [86].

### 3.5. Moisture

The moisture content of plant-based meat significantly impacted the fibre structure, texture, and flavour. Depending on the moisture content, one can categorize plant-based meat into low-moisture extrusion-cooked products and high-moisture extrusion-cooked ones. The former can be dried and stored for an extended period, while the latter has a higher moisture content and a shorter shelf life [87]. High moisture content ensures juiciness in meat-like products. Moreover, it can reduce viscosity, provide plasticity, generate heat of vaporization, and act as a solvent for reactions [88]. Most plant-based meat products, such as burger patties and sausages, have low moisture content. The production of high moisture extruded products presents challenges, such as higher production costs, complex processing, and a shorter shelf life.

### 3.6. Additives

Plant-based meat products are supplemented with additives to improve their quality, extend their shelf life, or increase their nutritional content [89] (Figure 3I,II). Guar gum, methylcellulose, and carrageenan improved the texture and structure of plant-based meat products due to the polyol structure and negatively charged groups binding to water through hydrogen bonding and ionic dipole interactions. This structure thereby increased the thickness and consistency of the meat-like product. It formed protein–polysaccharide mixtures, which increased water absorption and contributed to the formation of an anisotropic fibre structure [90]. In addition, their use in plant-based beverages can alter the texture or mouthfeel by delaying the gravitational separation of fat droplets (emulsification) or dense insoluble matter (precipitation) [61]. Adding colourants to plant-based products provides colour and increases consumer acceptability, while flavourings enhance the flavour and mouthfeel. Antioxidants and antimicrobials can prevent product oxidation and discolouration and improve the shelf life [4]. Extrusion processing technology can promote the application of gums as thickeners and stabilizers in the food industry [91]. The phenolics in antioxidants are thermally unstable compounds, and extrusion processing technology and heat treatment processing reduce the phenolics in vegetable protein-based products. The high temperature decomposes phenolics, or they polymerize [92]. At present, there is a need for a comprehensive investigation into the impact of extrusion processing technology on additives. This is essential for preventing fat rancidity and protein oxidation through the use of antioxidants and antimicrobial agents. By preventing oxidation and discolouration, the shelf life of plant-based meat can be extended [4]. The total polyphenol content and antioxidant properties of the meat analogue products were higher in comparison with the conventional meat products [93]. The soybeans with phenolic compounds demonstrated high antioxidant properties. Plant-derived antioxidant compounds are a chemically heterogeneous group. In the meat industry, product ingredients commonly include polyphenols (flavonols and anthocyanins) and essential oils (especially terpenoids). Soybeans contain polyphenols, such as isoflavones, lignans, phenolic acids, anthocyanins, tannins, and stilbenes. Food formulation utilizes phenolic chemicals due to their antioxidant properties and their effects against age-related diseases.

Extrusion processing technology allows gums to be thickeners and stabilizers in the food industry [91]. Phenolic compounds are thermally unstable. Thus, extrusion processing and heat treatments can reduce phenolic compounds in plant-based meat due to the degradation or polymerization with other components [92].

## 4. Challenges and Prospectives

Plant protein is a promising alternative to animal products in daily diets due to its unique suitability, wide availability, and protein bioavailability. Various processing methods maintain protein quality, quantity, palatability, taste, and good looks. There are existing challenges to plant-original protein processing in the global market. (1) One challenge lies in the non-environmentally friendly processing of plant-based protein, resulting in high costs and energy consumption due to low efficiency and leading to significant pollution. To tackle this issue, utilising low-cost plant resources and achieving high extraction efficiency is recommended. Currently, sourcing plant-based raw materials predominantly involves utilizing agricultural by-products, such as soybean and peanut meal. However, these limited sources cannot meet the large and diverse demands for plant-based protein. A comprehensive consideration for industrial-scale extraction techniques for plant proteins includes protein extraction efficiency, quality and quantity, low energy consumption, and environmental friendliness. For example, alkaline for protein extraction has imposed a significant burden on the environment, thus requesting more green, environmentally friendly, and sustainable regent. (2) A second challenge lies in the mass structure of plant-based proteins, requesting regulation of structures. Extrusion processing can achieve fibre-like structural reorganization of plant proteins but with imprecise control. The reason could be plant proteins’ complex composition and conformational variability, depending on processing temperature and time. Infrared spectroscopy and Raman spectroscopy are currently available for monitoring extrusion processes. The trend for monitoring techniques could be automation and personalization. For example, the 3D printing technique allows the comprehensive monitoring and precise control of the extrusion process. (3) The third one is the lack of evaluation and assessment of plant-based alternative protein products’ nutritional and health effects. Few works focused on plant-based protein products besides the available animal meat evaluation and assessment. For example, the preferred concentration of essential amino acids in plant-based proteins is not monitored and evaluated. Therefore, it is essential not to overlook the nutritional assessment of plant-based alternative protein products.

In the future, plant-based protein products will be an irreversible trend. According to a Food Marketing Institute (FMI) survey, taste is one of the key factors influencing consumer acceptance of plant-based alternative protein products. Data from the research agency Mintel also reveals that 53% of consumers hope that plant-based protein products taste indistinguishable from meat, indicating their desire for plant-based products to provide a similar taste experience to traditional meat products. Researchers have made numerous attempts to achieve colour, flavour, and texture in plant-based meat substitutes that closely resemble actual meat products. Potential research directions for future breakthroughs may encompass the following aspects. Firstly, compression techniques represent one of the most efficient and environmentally friendly methods for processing plant-based protein as a substitute. However, they fail to attain meat’s highly organized texture and water-binding capacity, thus falling short of genuinely replicating the authentic sensory experience of consuming meat. Nevertheless, novel approaches need to be explored to achieve comparable acceptance levels regarding the taste and texture of meat products. Secondly, emerging techniques such as high-pressure processing, ultrasound, microwave, ohmic heating, and irradiation offer opportunities to obtain a wider range of components, quality, and nutritional attributes for plant-based protein substitutes. Thirdly, conducting nutritional evaluations and assessing health effects will provide valuable feedback for optimizing and upgrading extraction and processing methodologies, thereby supporting the development of plant-based protein products for precision nutrition.

## Figures and Tables

**Figure 1 foods-12-04180-f001:**
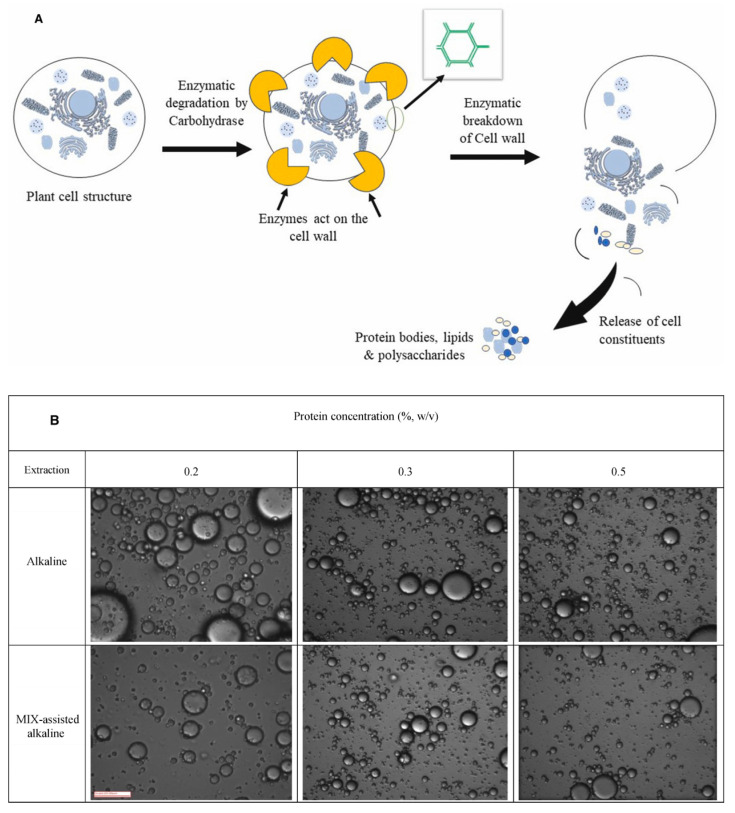
(**A**) Enzymatic breakdown of the plant cell wall during the extraction of proteins with the help of enzymes is shown schematically. (**B**) Creating emulsions with varying protein concentrations using a combination of alkaline extraction and carbohydrase enzymes [14,17].

**Figure 3 foods-12-04180-f003:**
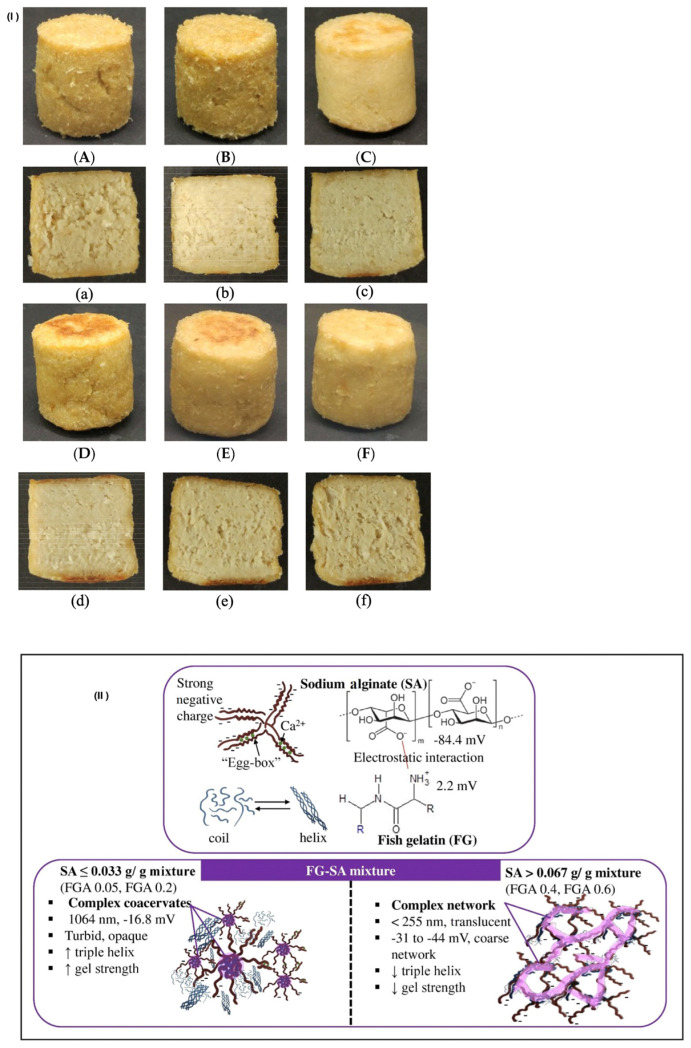
(**I**) The meat analogue’s exterior (**A**–**F**) and interior (**a**–**f**) appearance in six various liquid additions. Water is represented by (**A**,**a**); water and SPI by (**B**,**b**); canola oil by (**C**,**c**); canola oil and lecithin by (**D,d**); O/W emulsion by (**E**,**e**); and water and canola oil by (**F**,**f**). (**II**) The physicochemical and structural changes caused by a mixture of fish gelatin and sodium alginate are explained by the hypothesized mechanism [82,92].

## Data Availability

Data is contained within the article.

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
