# Peer review of "Plant-Based Meat Proteins: Processing, Nutrition Composition, and Future Prospects"

_foods, 2023, doi:10.3390/foods12224180_

Round 1
Reviewer 1 Report
Comments and Suggestions for Authors
This review manuscript is dealing with role of nutrients and changes in the nutrients during plant protein product processing to produce plant-based meat and dairy. This manuscript has an interesting manuscript and can be very useful for the food industry especially for those who are working on the plant-based functional foods.
Adding a table containing the previous studies related to the subject of this review article can improve the quality of the manuscript.
FAO: The Food and Agriculture Organization of the United Nations
L32-36 need a ref.
The introduction part in focused on plat meat. What about plant-based dairy which you also mentioned in the title?
Please add some previous studies using plant proteins to develop plant meat and dairy into the introduction part of the manuscript.
The aim of this review is not very clear to me. The aim of the manuscript should be explained in a better way in the introduction part.
L77: what is SDS?
The isoelectric point is 4.5 for all of the plant proteins?
L89: which proteins?
L94: what are the most common enzymes using for protein extraction?
I think that the results related to the extraction of plant proteins should be presented in a table containing their methods and results (for example their protein extraction efficiency). This table will be very useful.
L169: It will
One of the most common limitation for the plant protein is their poor techno-functional property such as solubility, foaming, and emulsifying ability. I think that the author should add a section to their review about the methods (such as fibrillation, enzymatic hydrolysis, pH-shifting and …) using for improving the functional properties of the plant proteins aimed to be used for the production of plant meat and dairy.
The 3rd part of the manuscript (nutrition in processing) is not written very well and its quality needs a huge improvement because it seems that this part is the most important part of this manuscript. I think that the author should also speak about the antioxidant activity of the plant-based products in this part because the oxidation process is a main process for the deterioration of the meat product quality.
Author Response
Dear reviewer, Thank you for your comments. We made comprehensive revision according to your comments, point by point. Please see the attachment.

Reviewer 2 Report
Comments and Suggestions for Authors
The authors report a literature review on nutritional components role during plant-based milk processing.
While the topic of the review is interesting, the structure and organization of the review by the authors is poor and inconsistent. Various aspects in the drafting of the text do not appear adequate for a review, such as, for example, the sentence in the introduction (lines 57-60) which indicates the title, the keywords and the search engines used.
The literature cited, consisting of 77 references, appears furthermore confused, and is based on method categorization integrated with processing role on nutritional components. Therefore, it should be added a general reference table summarizing main types of processing methodology.
Throughout the paper, the arguments of the authors appear confused and the different methodologies and food components are defined with a poor textbook language.
Another weakness, in the opinion of this reviewer, lies in the fact that authors do not provide adequate references to support the claim that general reviews exist (lines 48-50). However, the major criticality in the methodological framework of the review consists in the inconsistency of the title with the content of the text: while on the one hand the authors expose in the introduction (lines 53-54) the lack of comprehensive reviews on plant-based milk, on the other hand, the text contains papers for the vast majority based on plant-based proteins intended to obtain meat substitutes. So, it is not clear whether the review is on milk or products in general, because the cited literature seems not to concern milk.
In conclusion, it is not clear how the review can provide the scientific basis for future research on milk (lines 68-70). Indeed, in the paragraph Challenges and Prospectives (line 357) only arguments in favour and against this or that methodology are set out.
Overall comment
The review topic shows interest for the academic community, but the review needs at least a major revision to clarify the points addressed above. In its current form, the content is not acceptable; it is therefore recommended a total critical rewrite before being considered again for publication in the journal.
Author Response

(The authors gave the same response as above.)

Reviewer 3 Report
Comments and Suggestions for Authors
Totally, the research has not been well designed. Furthermore, the English is not good enough for publication and needs to revise. The quality of the work is so low! there isnt any figures and tables to summerzie the recent studies related to this topic! Authors wrote this article like a book chapter!
Comments on the Quality of English LanguageExtensive editing of English language required
Author Response

(The authors gave the same response as above.)

Reviewer 4 Report
Comments and Suggestions for Authors
The title of the review does not reflect the content of the text. There is not enough explanation about Dairy. In such studies, tables and figures should have been included in the manuscript, as in references 4 and 5. Tables and figures increase the reader's interest in the article. On the other hand, the explanations given in Line-57 and 61 are not necessary. The abbreviation SDS should be expanded on Line-77.
In short, the review should be enriched with figures and tables and the manuscript should be made more fluent. Once prepared in this way, it can be re-evaluated.
Author Response

(The authors gave the same response as above.)

Round 2
Reviewer 1 Report
Comments and Suggestions for Authors
the comments are applied. The quality of manuscript was improved.
Author Response
Thank you very much for taking the time to review this manuscript.
Reviewer 2 Report
Comments and Suggestions for Authors
The authors proceeded to an in-depth revision of the review, filling some of the gaps highlighted in the first submission. Furthermore, the authors proceeded to reorganize cited references with the insertion of a clear summary table.
Also, regarding definitions and language, there has been an improvement compared to the first version of the manuscript.
In this reviewer's opinion, the authors followed all suggestions and recommendations from the first review.
The manuscript requires, due to the expository language which does not appear completely linear, a revision by a native English speaker.
The authors proceeded to an in-depth revision of the review, filling some of the gaps highlighted in the first submission. Furthermore, the authors proceeded to reorganize cited references with the insertion of a clear summary table.
Also, regarding definitions and language, there has been an improvement compared to the first version of the manuscript.
In this reviewer's opinion, the authors followed all suggestions and recommendations from the first review; therefore, in this version the manuscript can now be taken into consideration.
Author Response
Thank you very much for taking the time to review this manuscript. The revision was polished by a native speaker.
Reviewer 3 Report
Comments and Suggestions for Authors
The manuscript does not reach the required quality standard
Comments on the Quality of English Languagepoor
Author Response
Thank you for your comments. The revision was polished a native speaker and revised according to the valuable comment.
Reviewer 4 Report
Comments and Suggestions for Authors
The article can be published. However, the article should be redesigned according to the journal format.
Author Response
Thank you for your comments. The manuscript could be redesigned by a publishing editor group. The corrections in the revised manuscript have been highlighted in red.